# Heparin–Superparamagnetic Iron Oxide Nanoparticles for Theranostic Applications

**DOI:** 10.3390/molecules27207116

**Published:** 2022-10-21

**Authors:** Nicolò Massironi, Miriam Colombo, Cesare Cosentino, Luisa Fiandra, Michele Mauri, Yasmina Kayal, Filippo Testa, Giangiacomo Torri, Elena Urso, Elena Vismara, Israel Vlodavsky

**Affiliations:** 1Department of Chemistry, Materials and Chemical Engineering “Giulio Natta”, Politecnico di Milano, 20133 Milan, Italy; 2Department of Biotechnology and Biosciences, University of Milano Bicocca, 20126 Milan, Italy; 3Istituto di Ricerche Chimiche e Biochimiche “Giuliana Ronzoni”, 20133 Milan, Italy; 4Department of Materials Science, University of Milano Bicocca, 20125 Milan, Italy; 5Rappaport Faculty of Medicine, Israel Institute of Technology, Haifa 2611001, Israel

**Keywords:** superparamagnetic iron oxide nanoparticles (SPION), heparin, heparanase, theranostic, paclitaxel, dopamine, toxicity, metastasis

## Abstract

In this study, superparamagnetic iron oxide nanoparticles (SPIONs) were engineered with an organic coating composed of low molecular weight heparin (LMWH) and bovine serum albumin (BSA), providing heparin-based nanoparticle systems (LMWH@SPIONs). The purpose was to merge the properties of the heparin skeleton and an inorganic core to build up a targeted theranostic nanosystem, which was eventually enhanced by loading a chemotherapeutic agent. Iron oxide cores were prepared via the co-precipitation of iron salts in an alkaline environment and oleic acid (OA) capping. Dopamine (DA) was covalently linked to BSA and LMWH by amide linkages via carbodiimide coupling. The following ligand exchange reaction between the DA-BSA/DA-LMWH and OA was conducted in a biphasic system composed of water and hexane, affording LMWH@SPIONs stabilized in water by polystyrene sulfonate (PSS). Their size and morphology were investigated via dynamic light scattering (DLS) and transmission electron microscopy (TEM), respectively. The LMWH@SPIONs’ cytotoxicity was tested, showing marginal or no toxicity for samples prepared with PSS at concentrations of 50 µg/mL. Their inhibitory activity on the heparanase enzyme was measured, showing an effective inhibition at concentrations comparable to G4000 (*N*-desulfo-*N*-acetyl heparin, a non-anticoagulant and antiheparanase heparin derivative; Roneparstat). The LMWH@SPION encapsulation of paclitaxel (PTX) enhanced the antitumor effect of this chemotherapeutic on breast cancer cells, likely due to an improved internalization of the nanoformulated drug with respect to the free molecule. Lastly, time-domain NMR (TD-NMR) experiments were conducted on LMWH@SPIONs obtaining relaxivity values within the same order of magnitude as currently used commercial contrast agents.

## 1. Introduction

Cancer, the second leading cause of death worldwide, is a group diseases in which abnormal cells divide without control and can invade nearby tissues. As cancer cells can spread to other parts of the body through the blood and lymph systems, it is not enough to identify and nurse the primary cancer, but it is necessary to detect metastasis early. Due to cancer’s nature, translational research on the cellular and molecular components of the cancer microenvironment has important implications for prevention, diagnosis, and treatment, the main tools that can combat any type of cancer [1].

Nowadays, contrast-enhanced magnetic resonance imaging (MRI) is one of the most effective methods of cancer and metastasis detection [2]. Superparamagnetic iron oxide nanoparticles (SPIONs) are the base of contrast agents such as Ferumoxide and Ferucarbotan, examples of commercialized formulations that are widely used as T2-weighted contrast agents (negative agent) and are rapidly replacing gadolinium-based contrast agents because of the risk of nephrogenic systemic fibrosis (NSF) [3,4,5]. Molecular imaging is one of the most promising applications of SPIONs, and various applications have been evaluated in vitro and in animal experiments [6].

As early diagnosis and focused therapy are two faces of the same medal, the theranostic approach of merging diagnosis and therapy could be the crucial tool against cancer [7]. The manipulation of SPIONs as theranostic agents has attained growing consideration, owing not only to their capability in MRI for diagnosis, but also in magnetic targeting, hyperthermia treatments, and drug loading or controlled release [8,9]. To intensify the diagnostic and therapeutic efficacy, SPIONs have been ornamented or functionalized with a range of materials. In addition, appropriate surface modifications impart SPIONs with better biocompatibility and stability and long blood circulation times. Preferentially synthetic and natural polymers have been exploited as coatings to ensure their colloidal stability and good dispersibility In biological fluids [10]. The choice of SPIONs as theranostic agents is also supported by their tendency to accumulate in tumor tissues due to the enhanced permeability and retention effect (EPR), granting passive specificity [11]. The challenges associated with the penetration of nanoparticles across cell and tissue barriers were recently reviewed [12]. An effective cancer therapy requires both passive and active forms of targeting, along with the consistent knowledge of various physiologic barriers to achieve drug-targeted delivery. Since 2010, nanocarriers for anticancer drug delivery have been endowed with active tumor targeting ability [13]. In addition to passive targeting, specific surface modifications can also be designed to endow SPIONs of cancer with active targeting ability [14,15].

The extracellular matrix (ECM) is a structural framework that has important physiological functions, which include maintaining tissue structure and integrity, serving as a barrier to invading pathogens, regulating cell differentiation and proliferation, and acting as a reservoir for bioactive molecules [16]. This cellular scaffold is made up of various macromolecules, including heparan sulfate (HS) and proteoglycans (HSPGs), the remodeling of which is important for physiological and pathological processes. HS interacts with a myriad of proteins in the ECM, facilitating diverse processes including growth factor signaling, cell–ECM adhesion, epithelial barrier formation, and endocytosis.

The turnover of HS is tightly regulated by heparanase (Hpa1), an endoglucuronidase enzyme capable of cleaving HS chains [17,18]. Hpa1 acts as a modulator of HSPGs and is a pivotal player in creating a permissive environment for tumor progression and inflammatory responses. Importantly, Hpa1 overexpression is associated with shorter event-free survival in cancer patients [19,20]. This and other clinical and experimental evidence has put forward the notion that Hpa1 is a drug target in cancer, inflammation, and other diseases, encouraging the development of Hpa1-targeted drugs [21,22,23].

Among the charged polysaccharides, heparin was reported to be a promising anticancer agent in patients with multiple metastases and with an increased risk of thromboembolism [24,25], due to its inhibitory effects on heparanase and P and L selectins [26,27]. Heparin, exclusively produced by mast cells, is a more sophisticated and sulfated form of HS. Early studies evidenced HS’s higher charged congener unfractionated heparin (UFH) as an efficient heparanase inhibitor and as a substrate [28,29,30,31]. Early studies showed effective heparanase inhibition by UFH, even if some UFH sequences can be recognized and cleaved by heparanase [32]. However, its unwanted anticoagulant activity hampered its safe use as an inhibitor of heparanase. The following investigations were mainly oriented to identify non-anticoagulant heparin derivatives to overcome UFH’s anticoagulation and bleeding side effects, as well as to improve its pharmacokinetics and bioavailability [23,33]. The elimination or at least the reduction in anticoagulant activity can be obtained through different types of chemical modifications of the structure of heparin, such as oversulfation, partial desulfation, a reduction in MW, or selective modifications of the residues at the antithrombin binding site, representing the major determinant of anticoagulant activity. The clinical use of low molecular weight heparins (LMWHs) in oncology was approved to prevent venous thromboembolism, as well as for their better pharmacokinetic and pharmacodynamics properties compared with UFH [25].

Given the above considerations, the present study aims to build up LMWH@SPIONs, a type of engineered superparamagnetic iron oxide nanoparticles (SPIONs; see Figure 1), whereby LMWH was chosen to endow the SPIONs with heparanase inhibition. The LMWH@SPION synthesis took inspiration from our previous paper by substituting hyaluronic acid (HA) with LMHWs in the oleic acid (OA) ligand exchange reaction. In addition, a SPION coating with bovine serum albumin (BSA) protein was used to modulate their biocompatibility and to reduce their toxicity [34]. Dopamine (DA) was chosen as a spacer between the SPIONs and LMWH/BSA, due to the catechol and amino groups. DA was firstly covalently linked to the LMWHs and BSA via amide linkage and carbodiimide coupling. Importantly, the DA–LMWH coupling requires the preservation of the LMWH-Hpa1 binding site sequences.

The following ligand exchange reaction between DA-LMWH/DA-BSA and OA, mediated by the catechol group, was conducted not only in the presence of cetrimonium bromide (CTAB) as a surfactant [34], but also in the presence of PSS, to face the drawback of CTAB’s toxicity.

The choice of LMWH was associated with the aim of chemotherapeutic drug loading, which could lead to reduced chemotherapy side effects, resulting in a lower clinical and economic burden [14,15]. We selected taxol (paclitaxel = PTX) as a chemotherapeutic agent that is widely used to treat different types of cancers [35]. The aim of the inclusion of PTX within LMWH@SPIONs is to improve the water solubility, pharmacokinetics, and biocompatibility.

## 2. Results

DA-LMWH was prepared as detailed in Section 4.1.

### 2.1. DA-LMWH NMR Characterization

DA-LMWH NMR studies have demonstrated the occurrence of the linkage between LMWH and DA, but also the impossibility of avoiding the direct condensation of LMWH-EDC, whereby one EDC unit was found for every seven LMWH disaccharides. Relevant information can be recovered about the LMWH sites involved in the DA linkage. In fact, due to the natural complexity of LMWH, the reaction can take place on the three different uronic acids, which are distributed along the polymer chain in different sequences. This distribution results in modified structural sequences with relevant effects on the chemical shifts of the NMR signals. For the NMR testing, two-dimensional homo- and heteronuclear single- and multiple-correlation techniques were used. The homonuclear correlation spectra (see Appendix A) allow the detection of variations in the chemical shifts of the protons in position 2 of the Ido2S that cannot be explained by 2-*O*-desulfation, as well as the permanence of signals due to EDC. From the distribution of the anomeric signals, it is possible to observe how the signals of non-sulfated uronic acids (glucuronic and iduronic acid) do not undergo variations in their number and chemical shifts (see Appendix A). Thus, we can argue that the condensation is selective for iduronic 2-*O*-sulphate signals. This result also that suggests DA-LMWH can inhibit heparanase because the specific sequence glucuronic acid-*N*-Acetyl glucosamine involved remains unchanged. On the other hand, the iduronic 2-*O*-sulphate signal of heparin (5.23 ppm), following the reaction, is subdivided into at least four components, indicating its involvement in the synthesis process of DA-LMWH. Since the derivatization reaction is aimed at the formation of amide bonds, and since the heteronuclear correlation spectrum does not allow one to observe the quaternary carbons, it is necessary to associate this spectrum with the long-range correlation spectrum (HMBC), which allows the identification of the correlations of quaternary carbons with protons at a multi-bond distance.

In the area of the aromatic signals of DA (Figure 1A), we observed the correlation of proton signals (^1^H: 6.7–7.1 ppm) with the quaternary aromatic carbons and with their aliphatic signals (^13^C: 130–150 ppm; ^1^H: 2.6–3.5 ppm), as well as with the signals of the heparin chain (^1^H: 3.0–3.5 ppm range). In Figure 1B, the one-dimensional ^13^C spectrum shows the appearance of new signals at 173.2 and 174.8 ppm, compatible with the formation of amide bonds, and whose correlation leads to the signals of protons in H5 of the residues of I_2S and 2 OH_. Regarding the persistence of the EDC signals, a correlation is observed between the C=O of uronic acid (176.9 ppm) and the CH_2_ signals in alpha to the tertiary amine of the EDC (^1^H/^13^C: 2.4/39 ppm). The presence of salts between heparin and the reaction by-product 1-(3-(dimethylamino) propyl)-3-ethylurea is highlighted by the correlations between the urea’s C=O at 163 and 159 ppm and the H2 signals of NSO_3_ glucosamine at 3.11, 3.20, and 3.32 ppm.

Due to the complexity of the spectra of the samples and the difficulty encountered in the procedures for their purification, we carried out diffusion-ordered spectroscopy (DOSY) measurements to discriminate the signals of the polymeric materials and those of the reactants or by-products. Figure 2 shows the type of spectrum obtained from the DA-LMWH. The spectrum allows the identification of both the atoms belonging to the DA-derivatized structure of the heparin chains and the free ones present in the solution. The signals at 2.4 and 1.9 ppm have polymeric and non-polymeric components. EDC-traceable signals are an integral part of the polymer. For the DA-LMWH studied in Figure 2, the relationship between the signals suggests an insertion of 2–3 DA residues for every LMWH chain. EDC-traceable signals are an integral part of the polymer. Importantly, by double fitting the Monte Carlo method, the heparin shows two different MW adducts both linked to DA (data not reported).

### 2.2. LMWH@SPION SIZE (DLS), Z-Potential, Morphology (TEM), and ICP-OES

The LMWH@SPION preparation via OA ligand exchange is detailed below in Section 4.2. Before assessing the impact of the produced SPIONs on living cells, the size (zeta average) and charge (zeta potential) were measured via DLS (Table 1) (data not reported). A positive charge was detected for SPION1 and SPION2, while SPION3 and SPION4 were highly and slightly negative, respectively. SPION1 is highly positive and well dispersed due to CTAB’s positive charge. For SPION2, more heparin seems to reduce the efficiency of CTAB (acting as heparin counter ion). SPION3 was the least stable of the synthesized systems and tended to aggregate after a few weeks, requiring sonication to redisperse the sample. In the presence of the steric hindrance of BSA, PSS seems unable to stabilize SPION3 for a long time. On the contrary, in the absence of BSA, PSS is very efficient in stabilizing SPION4, which was indeed stable at a concentration of 1 mg/mL, as confirmed by the DLS measurements conducted after 90 days, showing close to no variations in terms of size. The small Z-Av is in agreement with the lack of BSA, while the negative charge fits well with the exposed heparin. Regarding CTRL1 and CTRL2, CTRL1’s positive charge is in perfect agreement with CTAB’s presence, while CTRL2 is quite neutral due to the BSA in the zwitterion form.

Figure 3 shows SPION1, SPION3, and SPION4’s TEM images. SPION1’s morphology in Figure 3A was confirmed to be almost perfectly spherical and quite regular, with no appreciable differences from previously synthesized HA-SPIONs [34]. The morphologies of SPION3 in Figure 3C and SPION4 in Figure 3E are quite similar to the TEM image of magnetite particles capped with poly(styrenesulfonate-co-maleate) [36]. In the presence of CTAB, the SPION1 iron cores looks to be well dispersed by the organic layer, and the diameter is in the range of 3–6 nm. In the presence of PSS, SPION3 and SPION4’s iron cores appear to be more aggregated and the diameter is in the range of 6–10 nm. This difference could be explained by the presence of CTAB inside SPION1 as a bulky heparin counter ion. Figure 3B,D,F show the SPIONs’ dispersibility.

ICP-OES analyses were conducted to obtain the weight percentages of iron and sulfur helpful for TD-NMR and heparanase inhibition assays (Table 2)

We assume a sulfation rate of 2.5 per disaccharide, as calculated via NMR analysis for the parent heparin and reported in Section 2.7. The % S content in the BSA is negligible in terms of relative weight, and 13% of a disaccharide’s weight is composed of sulfur. Starting from the % S content obtained via ICP, subtracting the PSS’s contribution and multiplying this value by 7.5 (the latter being the disaccharide-to-sulfur weight ratio), we obtained values of 25% for SPION4 and 12.5% for SPION3.

### 2.3. Cytotoxicity

The safety of the different nanosystems was determined on HeLa cells incubated with increasing doses (1 to 50 μg/mL) of SPION1 and SPION2, in comparison to their LMWH-free DA-BSA@SPION CTAB^+^ control NP (CTRL1), and of the CTAB-free SPION3 in comparison with its CTAB-free control CTRL2.

Figure 4 shows that both SPION1 and SPION2 significantly impaired cell viability at doses higher than 25 μg/mL, but the same effect was observed also with CTRL1. Therefore, independently of the heparin content, the presence of the cationic detergent CTAB seems to play the main role in affecting cell viability.

The substitution of CTAB with PSS has proved much safer for cells. Indeed, no significant effect on the viability was detected with both SPION3 and the relative control CTRL2.

It had been already demonstrated that hybrid heparin/CaCO_3_/CaP nanoparticles were safe on HeLa cells at up to 500 µg/mL [37]. On the other hand, CTAB detergent has already been proven as a toxic agent for cells [38]. Moreover, we have to consider that the toxic effect observed with CTAB+ constructs could be also due to the positive net charge of this nanoparticle. Indeed, it is known that positively charged NPs present higher cell uptake and an enhanced impact on the cell membrane, resulting in increased cytotoxicity [39].

Hence, CTAB-free nanoformulations were selected for the following antitumor experiment on 4T1 cells.

### 2.4. PTX Loading

The PTX loading method is detailed below in in Section 4.5. The efficiency of the entrapment was verified by the visual disappearance of water-insoluble PTX crystals and by the lack of significant variations in terms of the hydrodynamic size. The DLS characterization was important because the lack of any peaks attributable to the presence of non-encapsulated PTX, which may not be visible to the naked eye, served as further confirmation that the SPIONs effectively stabilized the PTX against aggregation and crystallization (Table 3). In addition, the maintenance of Z-Av and Zeta-pot before and after loading (see Table 1) supports the successful encapsulation. The non-encapsulated PTX exhibited diameters of 2500 nm and higher, which were close to the upper detection limit of the DLS. These results suggest that the loading is quantitative.

### 2.5. Antitumor Activity of LMWH@SPION+PTX

Having demonstrated that the LMWH@SPION does not have cytotoxic effects on cancer cells, once the CTAB was removed, the combinatory effect of LMWH and PTX was detected on the 4T1 breast carcinoma cells. Indeed, a high expression of heparanase was detected in 4T1, and LMWH was shown to attenuate heparanase’s enzymatic activity, leading to the inhibition of cell migration and metastasis [21,40,41].

PTX was loaded into the nanosystems to evaluate the antitumor effect of PTX-containing SPIONs in comparison to the free drug. The effect on the cell viability was measured using the MTS assay after 48 h of incubation with increasing concentrations (from 1 to 50 µg/mL) of LMWH@SPIONs synthesized with or without DA-BSA, respectively known as SPION3 and SPION4, and loaded or not with PTX. The activity levels of the same SPIONs without LMWH (CTRL2) and free PTX were also evaluated as controls. Figure 5 shows that while PTX-free SPIONs, as well as CTRL2, did not influence the cell viability, PTX-containing SPION3 and SPION4 presented significant and dose-dependent cytotoxicity. Furthermore, when compared to free PTX tested in the same experimental conditions, the nanosystems displayed significantly higher antiproliferative activity on 4T1 cells (Figure 6). This result indicates that the SPIONs are effective in facilitating the cellular uptake of PTX, which is, therefore, able to reach the intracellular target in a higher amount. Hence, even though our nanoformulated heparin did not impair 4T1 cell viability, in agreement with previous evidence [42], the encapsulation of PTX into the heparin-based NPs enhanced the antitumor effect of this chemotherapeutic in breast cancer cells, likely due to the improved internalization of the nanoformulated drug with respect to the free molecule. It is feasible that this uptake exploits the interaction of heparin with the syndecan-1 receptor, which is physiologically involved in proheparanase endocytosis [43].

### 2.6. Inhibition of Heparanase’s Enzymatic Activity and ECM Degradation

We next investigated the ability of our LMWH@SPION preparations to abrogate heparanase of its native oligosaccharide substrates, using a colorimetric assay that detects the cleavage of the synthetic HS pentasaccharide fondaparinux [44,45]. The dose-dependent inhibition of the heparanase-mediated fondaparinux cleavage was exerted by both SPION3 (estimated IC50 = ~200 ng/mL) and SPION4 (estimated IC50 = ~70 ng/mL) (Figure 7). Notably, under the same conditions, our reference compound G4000 (=SST0001, Roneparstat = *N*-desulfo-*N*-acetyl glycolsplit heparin) [22] yielded an IC50 of 6.25 ng/mL. Given that LMWH respectively constitutes about 12.5% and 25% of the overall weights of SPION3 and SPION4 preparations, it appears that the LMWH structured within the newly synthesized superparamagnetic iron oxide nanoparticles retains its ability to inhibit heparanase’s enzymatic activity in a manner that is nearly similar to free LMWH.

Finally, we assessed the activity of the LMWH@SPION preparations against the heparanase-mediated degradation of bona fide HSPGs in an ECM environment. Cultured bovine corneal epithelial cells secrete a subendothelial basement-membrane-like material that is rich in proteoglycans, which can be radiolabeled via growth in the presence of ^35^SO_4_^2−^ [46]. Solubilized H^35^S chains released from H^35^SPGs via heparanase digestion can be resolved chromatographically, with heparanase inhibition leading to a corresponding loss of signal [47,48]. As demonstrated in Figure 8, both SPION3 and SPION4 caused nearly complete inhibition of the recombinant heparanase-mediated H35SPG cleavage, comparable to the reference molecule SST0001 [33]. The important conclusion is that the newly generated LMWH@SPION preparations are capable of inhibiting the degradation of HS, even when embedded in a naturally produced ECM that closely resembles basement membranes and the ECM in vivo. Years of research has indicated that compounds that effectively inhibit the enzyme in the ECM degradation assay are also effective in inhibiting cell invasion through the ECM in vitro and experimental metastases in preclinical animal models [19].

### 2.7. PTX Release from SPIONs

SPION3-PTX released the chemotherapeutic agent following a complex kinetic process, as shown in Appendix A, whereas SPION4-PTX released it following a zero-order kinetic process (Appendix A), as shown in the Appendix A and Section 4. The release profiles were quite different. Indeed, the presence of BSA plays an important role in determining the release kinetics of the SPION3+PTX system. Because of the high affinity of PTX to human plasma proteins of about 95%, BSA could act as a depot site for PTX, limiting the amount of drug that is available for release. This is also in agreement with the data on the antitumor activity of LMWH@SPION+PTX (see Section 2.6).

### 2.8. TD-NMR

Time=domain relaxation measurements via low field NMR were performed to check the paramagnetic iron persistency of the coated nanoparticles. All measurements were performed in triplicate; that is, the entire dilution process was performed from three different aliquots of the initial solution. We used slightly different dilution proportions for each series to better sample the concentration range. By measuring the T_1_ and T_2_ relaxation times at each concentration, the reciprocal R_1_ and R_2_ values were easily calculated and are plotted in Figure 9. There is good proportionality between the Fe content and relaxation parameters, indicating good dispersion. Slight deviations from linearity can be detected at the higher concentrations, but the used concentrations in the range of mg/mL are still close to the ideal dispersion.

The r_2_ values are over 80 s^−1^ Mm^−1^, a value comparable with many contrast agents in actual use [49]. While the spin–spin relaxation values are not substantially different between samples SPION3 and SPION4, the r_1_ is significantly higher for SPION4, decreasing the r_2_/r_1_ ratio. The resulting value, 5.9, is close to 5, the threshold for application as a T_1_ contrast agent. These kinds of agents are highly sought after because contrast agents based on r_1_ and r_2_ relaxation, respectively called light and dark contrast agents, are sensitive to different kinds of artifacts [50], but r_2_-based agents are far more common and easy to obtain. Basically, most magnetic particles can accelerate the transverse T_2_ relaxation, while obtaining good T_1_ relaxation with nanoparticles requires clever assemblies of magnetic centers with an external matrix [51]. The proton relaxation is strongly determined by the surface, which finely regulates the diffusion and retention and ultimately determines the timescale of the interaction between the nuclear spin and the magnetic cores [52]. We then presume that the removal of DA-BSA favors the surface interaction by increasing the accessibility of water to the surface.

## 3. Discussion

The synthesis of LMWH@SPIONs performed for this study follows in our opinion an innovative approach with respect to the previous studies. At the beginning of our interest in coating iron oxide nanoparticles with heparin, we chose the simplest approach—an electrostatic interaction between the preformed iron oxide nanoparticle and the highly negative heparin skeleton [53]. Heparin was detected on the SPION surfaces via FT-IR and the toluidine test. Among the various characterization techniques, ^1^H HR-MAS NMR was very interesting, which should, in principle, permit the detection of the carbohydrate connected to the nanoparticle’s surface [54]. Surprisingly enough, the NMR spectrum did not show the heparin, apparently in contrast with the FT-IR and toluidine test results. As NMR detects mobile structures, the lack of heparin detection can be explained by the drastic reduction in mobility due to the very strong connection to the metal oxide surface. As the final challenge was to take advantage of heparin’s biological properties, the absence of mobility hampered its capability due to the polymeric nature of heparin, which needs flexibility to act. In addition, the heparin coating did not guarantee the decrease in the nanoparticles’ intrinsic toxicity that was often envisaged [54].

The design of the LMWH@SPIONs was completely different, not only from our first electrostatic approach, but also from other well-established approaches. The most common synthetic methods use the co-precipitation of Fe(II) ions and Fe(III) ions in an aqueous basic solution to form SPIONs in the presence of heparin alone or associated with other polymers. In this way, the heparin/polymer becomes the organic SPION coating layer. For example, a SPION coating with heparin alone has been found to enhance the T2-weighted MR contrast for cell tracking in vivo [55]. Recently, ultracompact iron oxide nanoparticles with a monolayer coating of succinylated heparin have been proposed as a new class of renal-clearable and non-toxic T1 agents for high-field MRI. The increased porosity and hydrophilicity of the coating has been suggested to increase the water accessibility to the surface, resulting in increased magnetic properties [56]. Actually, heparin-based nanoparticles appear to be suitable for imaging, as they also possess good biocompatible characteristics, as outlined by the review published in 2018 [57]. Another exciting use found for heparin–poloxamer-coated SPIONs is entrapping the anticancer drug doxorubicin into the polymeric shell followed by controlled release [58].

Figure 1 shows how the core–shell approach to LMWH@SPIONs is hugely different from the simple layer coating approach. First, the DA acts as a spacer between the core and shell. Secondly, the DA’s adduct interaction with the core depends on the diol’s capability to complex the iron. This means that the assembly was designed according to the specific reactions and interactions of the different components. The explanation for the huge effort required for real chemistry is that the challenge of this paper is to endow SPIONs with specific properties of heparin, not simply to use the generic properties of polysaccharides. Thus, the structural characterization is fundamental, especially to identify which heparin groups are involved in the condensation with the DA. The NMR investigation (see Section 2.1) demonstrated the occurrence of the DA-LMWH linkage and the maintenance of the specific sequence of glucuronic acid-*N*-acetyl glucosamine involved in the heparanase interaction. This suggests that LMWH@SPION can inhibit heparanase, an important property that makes LMWH@SPION an anticancer and antimetastatic nanosystem. The results reported in Section 2.6 in Figure 7 confirmed the heparanase’s inhibition capability. In addition, the newly generated LMWH@SPION preparations are capable of inhibiting the degradation of HS, even when embedded in a naturally produced ECM that closely resembles basement membranes and the ECM in vivo (see Figure 8). These results confirm the maintenance of the heparin chain mobility and the iduronic acid’s conformational flexibility in the LMWH@SPIONs. As far as we know, the only approach to heparin-based SPIONs as heparanase inhibitors uses microwave-assisted synthesis [59]. This method produces very small SPIONs, with Z-average (nm) values of 55 to 29, with a very high heparin content range (70–90% weight), which appear in the TEM images as being rather different from the LMWH@SPIONs. In our opinion, the microwave action forms an organic thick multilayer that exposes on the surface a flexible heparin layer capable of interacting with heparanase.

The synthetic approach to LMWH@SPIONs needs to take into account the stability of their suspension and the toxicity aspects. CTAB as a surfactant and heparin counter ion guaranties the high stability of the suspension. The highly positive surface charge (see Section 2.2, Table 1) is due to its presence. However, CTAB is highly toxic. Due to the strong electrostatic interaction with heparin, it is impossible to reduce its amount. The many attempts to remove it failed, confirming the strong interaction with heparin, as stated for many years in the literature [60]. CTAB was successfully substituted by the non-toxic PSS, with the heparin counter ion being Na^+^, as confirmed by the negative surface charge.

Regarding the use of BSA as a component of the organic layer, at the beginning it was thought of as a biocompatible layer that can improve the stability [34]. Passing from HA to heparin, we found that the heparin layer is enough to stabilize the SPIONs. However, as detailed below, BSA plays a role in the last application of LMWH@SPIONs, as a drug delivery nanosystem obtained by entrapping PTX. PTX cannot be used alone, due to its very low solubility in water and low bioavailability. The LMWH@SPIONs suspensions were effective not only in solubilizing the PTX but also in entrapping it, as demonstrated by the maintenance of Z-Av and Zeta-pot before and after loading (see Table 1). As discussed in Section 2.5, in Figure 5 and Figure 6, the PTX-containing SPION3 and SPION4 presented significant and dose-dependent cytotoxicity. The SPIONs were effective in facilitating the cellular uptake of the PTX. Hence, the encapsulation of PTX into the heparin-based NPs enhanced the antitumor effect of this chemotherapeutic on breast cancer cells, likely due to an improved internalization of the nanoformulated drug in respect to the free molecules. The role of BSA is quite interesting, as both the kinetic studies (see Section 2.7) and the cytotoxicity results support the hypothesis that BSA is part of PTX. If confirmed, this result could open the way to long-term release induced by a BSA reservoir.

The results of these studies describe heparin–superparamagnetic iron oxide nanoparticles as a nanosystem suitable to be developed in nanomedicine, which is a non-toxic heparanase inhibitor capable of entrapping chemotherapy drugs and is a valid alternative to the currently used commercial contrast agents. This study opens the way to new applications of LMWH@SPIONs, maintaining the potential for contrast agent applications. The new applications require stable SPION suspensions in water, and PSS can be used to stabilize them, avoiding the use of the toxic CTAB.

## 4. Materials and Methods

Materials: The sodium nitrite (NaNO_2_), sodium borohydride (NaBH_4_), sodium acetate (NaOAc), Ethanol (EtOH), ferrous chloride (FeCl_2_), ferric chloride (FeCl_3_), oleic acid (OA), acetone, ammonia solution 30% (NH_4_OH), bovine serum albumin (BSA), dopamine (DA), *N*-hydroxysuccinimide (NHS), Ethyl-3-(3-dimethylaminopropyl)carbodiimide hydrochloride (EDC), 2-(*N*-morpholino)ethanesulfonic acid (MES), chloroform (CHCl_3_), hexane (Hex), cetrimonium bromide (CTAB), polystyrene sulfonate (PSS), CM sephadex C-25 resin, sodium chloride (NaCl), potassium chloride (KCl), sodium phosphate dibasic (Na_2_HPO_4_), potassium dihydrogenphosphate (K_2_HPO_4_), dialysis sacks (12,000 Da MWCO), and paclitaxel (PTX) were purchased from Sigma Aldrich (St. Louis, MO, USA). The Spectra/Por ^®^ 3 dialysis tubing (3500 Da MWCO) was purchased from Repligen Corporation (Kilbarry, Waterford, Ireland). The porcine mucosa unfractionated heparin API was kindly supplied by the Ronzoni Institute (Milan, Italy).

### 4.1. DA-LMWH and DA-BSA Preparation

Here, 60.4 mg of DA·HCl and 200 mg of LMWH (Dalteparin) (see Appendix A) were dissolved in 10 mL of MES buffer and magnetically stirred for 10 min. Next, 191.70 mg of 1-ethyl-3-(3-dimethylaminopropyl) carbodiimide chloride (EDC·HCl) and 57.50 mg of *N*-hydroxysuccinimide (NHS) were dissolved in the remaining 10 mL of MES buffer solution. The EDC buffered solution was then added dropwise to the DA/LMWH solution. The reaction ran for 6 h at 4 °C in the absence of light to avoid DA polymerization and was then dialyzed with a 3500 Da MWCO membrane against deionized water. The DA-LMWH purification from unreacted reagents and undesired by-products was conducted on a CM Sephadex C-25 device, a weak cation exchanger. The preparation of the DA-BSA was conducted in a similar manner, but with a different order of addition: 25 mL of 50 mM MES buffer was prepared and brought to pH = 6.2 with NaOH 0.5 M. Next, 250 mg of BSA, 160 mg of EDC·HCl, and 144 mg of NHS were added to the buffered solution and magnetically stirred for two hours. After two hours, 158 mg of DA·HCl was then added to the solution and the reaction proceeded under stirring for 6 h in the absence of light at 4 °C. Finally, the sample was dialyzed with a 12,500 Da MWCO membrane against deionized water.

### 4.2. LMWH@SPION Preparation by OA Ligand Exchange

SPION1 and SPION2 were synthesized in the presence of CTAB from 25 mg of OA@SPION and 7.5 mL of a DA-BSA solution (8 mg/mL), with 10 mg of DA-LMWH for SPION1 or 15 mg of DA-LMWH for SPION2 [34]. The DA-BSA characterization profiles are shown in the Appendix A. CTRL1 was synthesized as SPION1/2 in the absence of LMWH. A different procedure in which CTAB was replaced with PSS and CHCl_3_ with hexane [61] was pursued and led to the virtually non-toxic final nanosystems SPION3 and SPION4. SPION3: 25 mg OA@SPION was dispersed in 1.25 mL of hexane and 3 mL of EtOH was added to eliminate excess oleic acid (OA). The OA@SPIONs were recovered and dispersed in 20 mL of hexane. A 30 mL aqueous solution composed of 7.5 mL of DA-BSA (10 mg/mL in D.I. water) and 15 mg of DA-LMWH was brought to pH = 4.5 and added to the organic suspension. The exchange proceeded for 72 h under strong stirring and the aqueous phase turned dark brown. The phases were separated, the aqueous-phase pH was fixed to 7, and PSS at 0.30 mg/mL was added to the dispersion. The SPION3 suspension was ultrasonicated at 100 watts for 10 min. SPION4 was prepared in the same way as SPION3 without the addition of DA-BSA. CTRL2 was synthesized in the same way as SPION3 without the addition of DA-LMWH. The nanosystems were purified via dialysis (12,000 kDa MWCO). SPION1, SPION2, and SPION3 were filtered on a 200 nm regenerated cellulose filter prior to the cell viability and heparanase inhibition assays. The SPIONs’ FT-IR spectra demonstrated the occurrence of the ligand exchange (see Appendix A).

### 4.3. Cell Viability Assay

The HeLa cells were seeded on a 96-multiwell dish at a density of 1.4 × 10^4^ cells/well and grown for 24 h. Then, the cells were incubated with SPION1, SPION2, and SPION3 and the control SPIONs’ CTRL1 (with CTAB) and CTRL2 (without CTAB) at concentrations of 1, 10, 25, and 50 μg/mL. Following 24 h of incubation, the medium was refreshed, 20 µL of 3-(4,5-dimethylthiazol-2-yl)-5-(3-carboxymethoxyphenyl-2-(4-sulfophenyl)-2H-tetrazolium (MTS) stock solution (Promega, Milano, Italy) was added to each well, and the cells were incubated for 3 h at 37 °C.

Here, the 4T1 cells were seeded on a 96-multiwell dish at a density of 1.4 × 10^4^ cells/well and left to recover overnight. The cells were then incubated for 48 h with SPION3 and SPION4, both with and without PTX, at vector concentrations of 10 and 50 µg/mL (respectively equivalent to 0.13, 1.25, and 6 µg/mL of PTX). The heparin-free CTRL2 and free PTX were tested as controls at concentrations equivalent to those found in the heparin/PTX-containing samples. At the end of the incubation time, the medium was refreshed, MTS was added to each well, and the cells were incubated as described above.

For both cell lines, the 490 nm absorbance results were then measured with an EnSight™ Multimode Plate Reader (PerkinElmer, Waltham, MA, USA) and te cell viability was calculated by normalizing the absorbance of the treated samples against the level recorded in the untreated sample. The results are expressed as means ± standard errors of measurement (*n* = 4).

### 4.4. Heparanase Activity Assay

ECM degradation assay: The ECM was prepared via incubation with recombinant human heparanase in the presence and absence of the indicated concentrations of the inhibitory compound, as described in [46], using the same reaction mixture and procedure. The sulfate-labeled material eluted in peak I (fractions 3–10, just after the void volume) represents the nearly intact HS proteoglycan released from the ECM due to proteolytic activity residing in the ECM [47,48].

Colorimetric (Arixtra) assay: The assay was carried out in a 96-well plate in which the formation of the disaccharide product upon the cleavage of fondaparinux (Arixtra, GlaxoSmithKline, Brentford, UK) was estimated using WST-1 tetrazolium salt, as described earlier [45].

### 4.5. PTX Loading and Release

The PTX was included by adding 50 μL of EtOH 5 mg/mL to 3 mL of SPION3 and SPION4 in 1 mg/mL of aqueous suspension with vortexing for 72 h. The PTX was successfully included and surprisingly soluble at a concentration of 125 μg/mL (150 μM). The PTX release was detected using an HPLC-UV system after being submitted to a dialysis assay. The used column was a Hypersil BDS C18 column (250 × 4.6 mm, 5 µm) on a Knauer Smartline Pump 1000 device equipped with an LC 6-Port/3-Channel Injection Valve autosampler, vacuum degasser, and fraction collector (Knauer, Berlin, Germany).

### 4.6. Nuclear Magnetic Resonance Spectroscopy (NMR)

The NMR spectra were recorded at 30 °C using a 500 MHz Avance NEO spectrometer (Bruker, Billerica, MA, USA) equipped with a TCI cryoprobe. About 28 mg of the samples was dissolved in D2O (99.996%). The proton spectra were recorded with presaturation of the residual water signal with 8–16 scans. The heteronuclear single-quantum coherence (HSQC) spectra were obtained in phase sensitivity with 24 scans, while the heteronuclear multiple bond correlation spectra (HMBC) were obtained with 64 scans. The matrix size of the experiments was 1024_320 data points and was zero-filled to 4096_2048 via the application of a shifted (π/2) squared cosine function prior to a Fourier transformation. The diffusion-ordered spectroscopy (DOSY) spectra were acquired using 32 scans and a series of 16-spin echo spectra registered with a time domain of 16 K zero-filled to 64 K.

### 4.7. Matrix-Assisted Laser Desorption Ionization–Time of Flight (MALDI-TOF)

Mass spectrometry measurements were conducted using a UV-MALDI-TOF Autoflex III system (Bruker, Bremen, Germany) equipped with a 355 nm pulsed UV laser. The sample solutions prepared at the concentration of 5–10 mg/mL in water and were mixed at a 1:1 (*v*/*v*) ratio with the matrix solution composed of a saturated solution of sinapinic acid in 0.1% trifluoroacetic acid with acetonitrile at 2:1 (*v*/*v*). The spectra were recorded in linear mode and the positive polarity levels in the mass ranged from 10 to 150 kDa.

### 4.8. Fourier Transform Infrared Spectroscopy (FT-IR)

The solid phase FT-IR spectra of the powdered sample with infrared-grade KBr were generated using an Alpha spectrometer (Bruker, Bremen, Germany). The data were analyzed using OPUS software version 7.0 (Bruker, Bremen, Germany).

### 4.9. Transmission Electron Microscopy (TEM)

The samples for the transmission electron microscopy (TEM) analysis were prepared by depositing a drop of each aqueous dispersion on a 3 mm copper grid coated with a thin layer of amorphous carbon and letting it air-dry overnight at room temperature before the analysis. The grids were then observed under a Philips CM 200 field emission gun (FEG) and a TEM system operating at an accelerating voltage of 200 kV.

### 4.10. Dynamic Light Scattering (DLS)

The hydrodynamic diameter expressed as intensity (Z_av_) and zeta potential (ξ) values of the nanosystems were measured using the Zetasizer Nano ZS (Malvern, Worcestershire, UK) with a fixed 173° scattering angle and a 633 nm helium neon laser. The data were analyzed using Zetasizer software version 7.11 (Malvern, Worcestershire, UK). The temperature was set to 298 K. SPION1, SPION2, and SPION4 were collected from the bulk mixture after the ligand exchange procedure and directly submitted to DLS measurements. The SPION3 systems were passed through a 0.22 µm syringe filter, sonicated for 10 min, and then submitted to DLS measurements (3 series of 12 measurements).

### 4.11. Inductively Coupled Plasma Spectrometry (ICP)

Here, 10 mg of lyophilized SPION3 and SPION4 was mineralized with nitric acid (HNO_3_) and subsequently analyzed using an ICP spectrometry instrument (Perkin Elmer Optica 2300, Perkin Elmer, Milan, Italy).

### 4.12. TD-NMR

The measurements were performed on a Bruker minispec mq20 with 1.5 T magnetic field, corresponding to a 20 MHz proton resonance. Samples at different concentrations were obtained via successive dilutions of the stock solution with phosphate-buffered solution (PBS) to maintain a constant pH. The resulting dispersions were stable in the timescale of hours and sometimes days. The relaxometric parameters T1 and T2 were then measured at a constant temperature of 37 °C with saturation recovery and CPMG sequences, respectively [62].

## Data Availability

Not applicable.

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
