# Peer review of "Heparin–Superparamagnetic Iron Oxide Nanoparticles for Theranostic Applications"

_molecules, 2022, doi:10.3390/molecules27207116_

Round 1
Reviewer 1 Report
1. Section 1. Introduction.
(1)“The use of LMWH can not only lead to heparanase inhibition, but may also contribute to active targeting of the tumor mass by chemotherapeutic drugs“ Why dose Hpa1 degrade HSPG as a modulator while LMWH can inhibit heparanase? Why does LMWH contribute to active targeting of the tumor mass?
(2) Why was BSA used to form LMWH@SPIONs?
2. The sentence “DA-LMWH was prepared as detailed in Section 3, paragraph 3.1.” should be cancelled.
3. How to synthesize DA-BSA?
4. Fig.5 shows the serious aggregation of nanoparticles. The TEM images are suggested to re-perform.
5. The cell viability of SPION4 and LMWH should be examined.
6. Why can SPION load PTX? What is the loading content of PTX?
7. Figure 8 is not intact.
8. How to perform PTX release test?
9. The overall presentation should be improved.
Reviewer 2 Report
The manuscript is hard to read due to a huge amount irrelevan details. Most of the text in the "Results" section looks like experimental details and should be shifted to SI. In my opinion the maniscript should be largely revised in order to be able to assess its scientific significance.
Specific comments:
1. The text on NMR characterization of DA-LMWN looks redundant in “Results” section (2.1). I would recommend shifting part of the text to Supplementary Materials
2. The section 2.2 looks like an Experimental section but not “Results”
3. Figure 5 shows TEM images for different SPIONs. Why are the images for the same SPIONs presented at different magnification that do not add the information about morphology of NPs? Moreover, the resolution of TEM images is very low for calculating the size of NPs. I would recommend deleting or rewriting this paragraph as no valuable information can be gained from it.
4. Figure 8. I believe that some part of the Figure is absent.
5. I suggest authors to conduct a TGA analysis to determined a amount of LMWH at SPIONs to better characterize the NPs.
6. In section 2, line 128 authors wrote that “DA-LMWH was prepared as detailed in Section 3, paragraph 3.1.” Actually, I have found this experiment in section 4 (Materials and Methods). The numeration in the section should be corrected accordingly.
Round 2
Reviewer 1 Report
More correction of the paper is needed.
1. According to the references, LMWH can inhibit heparanase, which will be related to tumor growth, but it is not scientific to mention the active targeting effect of LMWH. Moreover, the paper didn’t provide the experimental data on animal model to confirm the active targeting effect. Therefore, this point should be reconsidered.
2. The action of BSA should be explained in Section 1. Introduction.
3. Figure 3 only shows one particle in each photo. It’s not enough to display the sample morphology. Each photo should display several nanoparticles with good dispersibility.
4. In Table 1, please explain the meaning of “+”,”++” and “-“.
5. In Figure 8, what did the yellow and grey curves present?
6. The method to load PTX should be described in Section 4 but not in Section 2.4.
Reviewer 2 Report
I still don't understand what is the difference in TEM (Figure 3) between images A and B, C and D, E and F for SPIONs. This drawing remains confusing; why do the authors present these images where B duplicates A, D duplicates C, and F is a repetition of E at the same scale bar?
Another concern is with SPION magnetic separation for TGA measurements to determine the amount of organic ligand on the SPION surface. The authors stated in their response that "systems generally have high colloidal stability and cannot be separated by centrifugation or magnetic decantation." In this regard, the question arose: if the SPION system is not susceptible to the magnetic field in the case of magnetic separation, how can this system work in magnetic resonance therapy?
I suggest authors to conduct the magnetic mesurements of the synthesized SPIONs to elicidate their magnetic behavior.
